# *ppargc1a* controls nephron segmentation during zebrafish embryonic kidney ontogeny

Joseph M Chambers[1,2,3], Shahram Jevin Poureetezadi[1,2,3], Amanda Addiego[1,2,3], Manuela Lahne[1,2,3], Rebecca A Wingert[1,2,3]*

[1]Department of Biological Sciences, University of Notre Dame, Indiana, United States; [2]Center for Stem Cells and Regenerative Medicine, University of Notre Dame, Indiana, United States; [3]Center for Zebrafish Research, University of Notre Dame, Indiana, United States

**Abstract** Nephron segmentation involves a concert of genetic and molecular signals that are not fully understood. Through a chemical screen, we discovered that alteration of peroxisome proliferator-activated receptor (PPAR) signaling disrupts nephron segmentation in the zebrafish embryonic kidney (*Poureetezadi et al., 2016*). Here, we show that the PPAR co-activator *ppargc1a* directs renal progenitor fate. *ppargc1a* mutants form a small distal late (DL) segment and an expanded proximal straight tubule (PST) segment. *ppargc1a* promotes DL fate by regulating the transcription factor *tbx2b,* and restricts expression of the transcription factor *sim1a* to inhibit PST fate. Interestingly, *sim1a* restricts *ppargc1a* expression to promote the PST, and PST development is fully restored in *ppargc1a/sim1a*-deficient embryos, suggesting Ppargc1a and Sim1a counterbalance each other in an antagonistic fashion to delineate the PST segment boundary during nephrogenesis. Taken together, our data reveal new roles for Ppargc1a during development, which have implications for understanding renal birth defects.
DOI: https://doi.org/10.7554/eLife.40266.001

*For correspondence:
rwingert@nd.edu

**Competing interests:** The authors declare that no competing interests exist.

## Introduction

The vertebrate kidney develops from the intermediate mesoderm and can have two or three stages depending on the organism, where amphibians and fish develop a pronephros and mesonephros, while others like birds and mammals form these structures and also generate a metanephros (*Saxén and Sariola, 1987*). Roles of the kidney include regulation of osmolarity, fluid balance, and blood filtration. The kidney performs these tasks with functional units known as nephrons. Nephrons reabsorb or secrete precise amounts of essential molecules, ranging from amino acids to electrolytes, based on the dynamic physiological needs of the organism. Nephrons are divided into three main components: the filtration unit, a tubule, and a collecting duct (*Romagnani et al., 2013*). Nephron tubules are further subdivided into unique epithelial segments that perform the specialized tasks of reabsorption or secretion of discrete cargos. Ongoing advances have shed light on a number of the molecular pathways and gene expression signatures associated with nephron segment formation in kidney forms across species (*Desgrange and Cereghini, 2015*; *Lindström et al., 2015*; *Lindström et al., 2018*). Nevertheless, there are still many remaining questions about the genetic mechanisms that control segment fates and what specifies boundary formation between adjacent nephron segments.

The embryonic zebrafish pronephros is a tractable model to study the processes of nephron segmentation (*Gerlach and Wingert, 2013*). Completely segmented at just 24 hours post fertilization (hpf), the pronephros is composed of two bilateral nephrons that possess a conserved order and

arrangement of segment populations similar to other vertebrate nephrons (*Wingert and Davidson, 2008*). These domains include two proximal and two distal tubule segments: the proximal convoluted tubule (PCT), proximal straight tubule (PST), distal early (DE), and distal late (DL) (*Wingert et al., 2007*; *Wingert and Davidson, 2011*). Additionally, genetic studies in zebrafish are readily performed using reverse approaches like genome editing, knockdown or various overexpression techniques, as well as forward approaches like chemical genetics. Recently, we reported the results from a novel small molecule screen using the zebrafish pronephros as a segmentation model, which included the discovery that modulators of peroxisome proliferator-activated receptor (PPAR) signaling altered nephron segmentation (*Poureetezadi et al., 2016*). Until the present study, however, the functions of discrete PPAR signaling components during nephrogenesis have remained unknown.

PPAR family member peroxisome proliferator-activated receptor γ coactivator 1-alpha (Ppargc1a in zebrafish, PGC-1α in mammals) was discovered as a transcriptional coactivator for several nuclear hormone receptors such as PPAR alpha and gamma (PPARα, PPARγ), histone acetyltransferase steroid receptor coactivator 1 (SRC-1), and thyroid hormone receptor (*Puigserver et al., 1998*; *Puigserver et al., 1999*; *Wu et al., 1999*). PGC-1α serves diverse functions in various contexts not only as a transcriptional coactivator, but also by interactions with chromatin remodeling factors and RNA processing complexes (*Knutti and Kralli, 2001*; *Puigserver and Spiegelman, 2003*). PGC-1α is well known to regulate mitochondrial biogenesis and cellular metabolism (*Lynch et al., 2018*). Additionally, PGC-1α mediates the hepatocyte glucogenesis response to fasting (*Herzig et al., 2001*; *Yoon et al., 2001*), cardiac muscle and other slow-twitch muscle development (*Lin et al., 2002*; *Russell et al., 2004*), regulates angiogenesis (*Arany et al., 2008*; *Patten et al., 2012*; *Saint-Geniez et al., 2013*) as well as intestinal and skeletal stem cell fate during aging (*D'Errico et al., 2011*; *Yu et al., 2018*).

There has been an increasing appreciation for the roles of PGC-1α in adult renal physiology and disease, related in part to high metabolic demands of the kidney and the fact that it is the second most mitochondrial abundant organ (*Pagliarini et al., 2008*). PGC-1α is expressed in both the adult human kidney and the adult mouse kidney, specifically in the cortex and outer medulla (*Tran et al., 2011*; *Fagerberg et al., 2014*; *Tran et al., 2016*; *Casemayou et al., 2017*; *Han et al., 2017*). In experimental murine models of acute and chronic kidney disease, PGC-1α activity has been shown to mediate renoprotection in tubular cells, and deleterious outcomes have been associated with lowered or absent PGC-1α levels in various kidney injury models (*Lynch et al., 2018*). For example, in acute damage settings, PGC-1α expression is decreased and correlates with elevated fibrosis, whereas damage progression is attenuated when PGC-1α expression is induced (*Tran et al., 2011*; *Tran et al., 2016*; *Han et al., 2017*). Interestingly, *ppargc1a*/PGC-1α expression has also been annotated in many tissues of developing zebrafish and mice, including the kidney, but their roles in organogenesis events at these locations have not yet been fully ascertained (*Bertrand et al., 2007*; *Thisse and Thisse, 2008*; *Diez-Roux et al., 2011*; *Finger et al., 2017*). Most pertinent to the current study, the purpose(s) for the expression of *ppargc1a* transcripts in nascent nephrons has not been explored up until this point.

Here, we report the discovery that *ppargc1a* has essential roles during nephron segmentation in the zebrafish embryonic kidney. Our studies reveal that the spatiotemporal localization of *ppargc1a* transcripts in the developing intermediate mesoderm is highly dynamic, where expression throughout the renal progenitors becomes progressively localized to subdomains of the distal nephron segment precursors. Through loss-of-function studies, we show that *ppargc1a* is necessary for proper formation of two nephron segments, the DL and PST. Furthermore, our genetic studies demonstrate that *ppargc1a* influences the regionalized expression domains of two essential transcription factors, *T-box 2b* (*tbx2b*) and *SIM bHLH transcription factor 1a* (*sim1a*), which specify the DL and PST segments, respectively. We discovered that the PST segment boundary is established by an antagonistic relationship between *ppargc1a* and *sim1a*. Further, our data reveal that this opposing interaction constitutes a fascinating layer of redundancy with respect to other events that orchestrate nephron segmentation. Taken together, these studies divulge novel mechanisms that define nephron segment boundaries in the embryonic renal mesoderm. Our findings have implications for

understanding the basis of nephrogenesis in humans during normal development and congenital disorders affecting renal ontogeny as well.

## Results

### Bioactive small molecule chemical genetic screen reveals that alteration of PPAR signaling leads to changes in embryonic nephron segmentation

Chemical genetic screening is an efficient method used to employ the strengths of the zebrafish as a model organism to study a wide range of biological processes (*North et al., 2007*; *Garnaas et al., 2012*; *Nissim et al., 2014*; *Poureetezadi et al., 2014*). By applying different compounds to embryonic zebrafish, one is able to identify novel regulators in a high-throughput manner (*Poureetezadi and Wingert, 2016*). In a chemical genetic screen of known bioactive compounds, we identified novel regulators of zebrafish pronephros segmentation using a riboprobe cocktail to survey alternating tubule populations (*Poureetezadi et al., 2016*). One class of identified hits was compounds known to alter the activity of PPAR signaling (*Figure 1—figure supplement 1A,B*). For example, bezafibrate, a PPAR alpha agonist, was found to reduce the length of the PCT and DE tubule segments, suggesting alterations in processes such as the patterning, growth or cell turnover in the developing nephron (*Figure 1—figure supplement 1A,B*). Treatment with two PPAR gamma antagonists, BADGE and GW-9662, was associated with an increased DE; further, GW-9662 treatment was also scored as leading to a PCT segment increase (*Figure 1—figure supplement 1A,B*). These results similarly suggested that alterations in PPAR signaling could modulate nephron segmentation.

To further explore the PPAR pathway result, we collected wild-type (WT) zebrafish embryos and then treated them with dimethyl sulfoxide (DMSO) vehicle control or 150 µM bezafibrate/DMSO from the 5 hpf stage (approximately 50% epiboly) until the 28 somite stage (ss) when the nephron is fully segmented. After removing the drug, embryos were fixed and whole mount *in situ* hybridization (WISH) was performed to specifically assess formation of each individual nephron tubule segment. For this, we utilized riboprobes to detect transcripts encoding: *slc20a1a*, to mark the PCT; *trpm7*, to mark the PST; *slc12a1*, to mark the DE; and *slc12a3,* to mark the DL (*Wingert et al., 2007*). Embryos incubated with bezafibrate displayed a significantly increased length of the PST segment and a reduced DL segment compared to WT controls (*Figure 1A,B*). In contrast, there were no significant changes in the length of either the PCT or DE segments (*Figure 1—figure supplement 1C*). This set of phenotypes was present in the majority of embryos (*Figure 1—figure supplement 1D*). These findings indicated that emergence of the PST and DL segment populations can be modulated by changes in PPAR signaling, and suggested that some component(s) of the PPAR network might normally serve as renal regulators during nephrogenesis.

### *ppargc1a* is dynamically expressed in the developing zebrafish embryonic kidney

Intrigued by these results, we next sought to identify whether the expression of any PPAR signaling components would situate them as possible candidates for involvement in nephron development. We surveyed online expression repositories and published literature and found that transcripts encoding *ppargc1a/PGC-1α* have been detected in nascent nephrons within the developing zebrafish pronephros and the mouse metanephros (*Bertrand et al., 2007*; *Thisse and Thisse, 2008*; *Diez-Roux et al., 2011*; *Ai et al., 2017*; *Finger et al., 2017*). To further investigate this, we performed WISH on WT embryos and assessed the spatiotemporal patterns of *ppargc1a* expression throughout the stages of zebrafish pronephros ontogeny. *ppargc1a* transcripts were expressed in a pattern suggesting their presence in the entire populace of the developing intermediate mesoderm at the 8 ss, followed by a caudal restriction at the 20 ss before localizing to the distal segments at the 28 ss (*Figure 1C*, *Figure 1—figure supplement 2*). At the 30 hpf time point, weak expression levels of *ppargc1a* transcripts were detected in the proximal tubule as well (*Figure 1—figure supplement 2A*).

To confirm that *ppargc1a* was expressed in renal precursors, we employed double fluorescent WISH (FISH) and confocal imaging in WT embryos. At the 15 ss, *ppargc1a* transcripts were co-localized in the entire domain of cells that expressed transcripts encoding the intermediate mesoderm

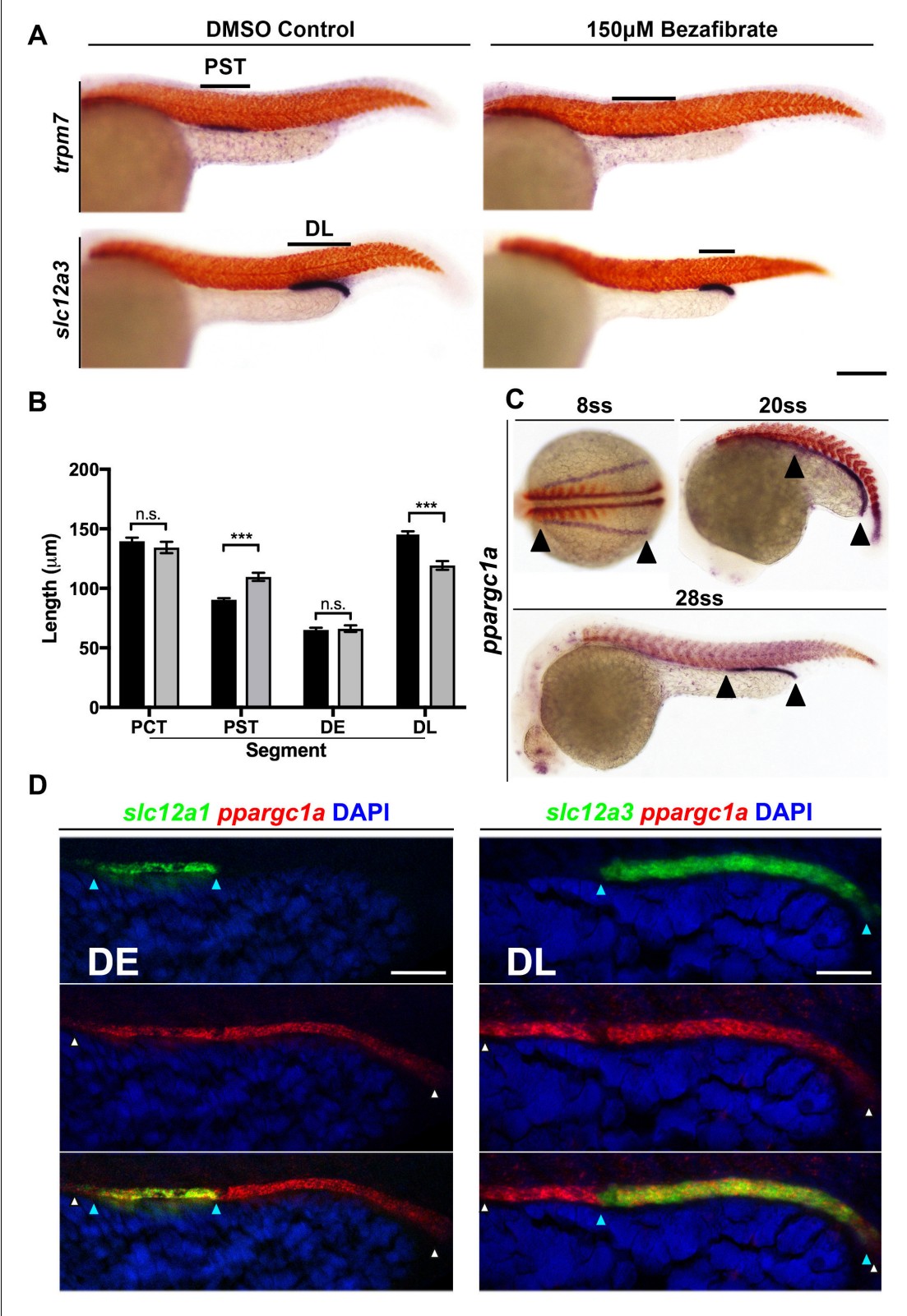

**Figure 1.** PPAR agonist bezafibrate alters zebrafish pronephros segmentation, and the PPAR coactivator *ppargc1a* exhibits a dynamic expression pattern in renal progenitors. (**A**) Double WISH at the 28 ss for the PST segment marker *trpm7* (top), and the DL segment marker *slc12a3* (bottom), with *smyhc* (red) to mark somites in DMSO control (left) and PPAR agonist, 150 µM Bezafibrate-treated (right) samples confirmed the initial hit from the chemical screen. Scale bar = 90 µm. (**B**) Absolute length measurements of the changes to pronephros segment lengths in bezafibrate treated (grey) and
*Figure 1 continued on next page*

*Figure 1 continued*

control samples (black). (C) Double WISH for *ppargc1a* (purple) expression at the 8 ss, 20 ss, and 28 ss with somites stained (red) (8 ss = *deltaC*, 20 ss and 28 ss = *smyhc*). (D) Double FISH at the 28 ss showing colocalization of *ppargc1a* (red) with *slc12a1* (distal early, left) and *slc12a3* (DL, right). Expression boundaries are indicated with blue (DE, left and DL, right) and white (*ppargc1a*) arrowheads. Scale bars = 35 μm. Data are represented as ±SD, significant by t test comparing the drug treatment to the DMSO vehicle control, n.s. = not significant, *** = p < 0.001.

DOI: https://doi.org/10.7554/eLife.40266.002

The following figure supplements are available for figure 1:

**Figure supplement 1.** Chemical genetics analysis of nephron development following exposure to PPAR pathway modulators.

DOI: https://doi.org/10.7554/eLife.40266.003

**Figure supplement 2.** *ppargc1a* mRNA transcripts are expressed throughout the intermediate mesoderm before restricting to the distal nephron segments.

DOI: https://doi.org/10.7554/eLife.40266.004

marker *paired box 2a* (*pax2a*) (*Figure 1—figure supplement 2B,C*) (*Krauss et al., 1991*; *Püschel et al., 1992*). By the 28 ss, *ppargc1a* transcripts were colocalized only in cells that expressed the distal segment markers *slc12a1* and *slc12a3*, indicating restriction to the DE and DL, respectively (*Figure 1D*). These data provide strong evidence that renal progenitors, followed by segment precursors and eventually differentiated distal segments, express *ppargc1a*. Based on the evidence that *ppargc1a* is dynamically expressed in the developing nephron, we hypothesized that it was involved in segment patterning.

## *ppargc1a* is necessary for proper formation of proximal and distal segment boundaries

To define whether *ppargc1a* is essential for nephrogenesis, we designed several parallel strategies to perform loss of function studies. The *ppargc1a* locus is comprised of a series of 12 exons (*Figure 2A*), and these encode a peptide that shares high sequence similarity, particularly in key functional domains, to mouse and human PGC-1α (*Figure 2—figure supplement 1*) (*Puigserver and Spiegelman, 2003*). Four distinct locations of the *ppargc1a* sequence were targeted for experimental manipulation (*Figure 2A*, *Figure 2—figure supplement 1*) in order to disrupt transcriptional processing or translation. First, we obtained a *ppargc1a* genetic knockout line (*ppargc1a$^{sa13186}$*), which encodes a T->A substitution located in exon seven that results in a premature STOP codon and eliminates a series of essential peptide domains (*Figure 2A*) (ZIRC - Eugene, Oregon; *Busch-Nentwich et al., 2013*). Sequencing confirmed the mutation and we developed a genotyping assay, which utilizes PCR amplification followed by NdeI restriction fragment length polymorphism digest analysis where the enzyme can cut the WT but not the mutant allele (*Figure 2—figure supplement 2D*). Second, we developed genetic models of *ppargc1a* deficiency using morpholinos (MOs). These included a translation blocking MO (MO1) (*Hanai et al., 2007*; *Bertrand et al., 2007*) and two splice blocking MOs (SB MO1, SB MO2) that we designed and subsequently validated through microinjection and RT-PCR studies in WT embryos (*Figure 2A*, *Figure 2—figure supplements 3* and *4*).

The *ppargc1a* mutant and knockdown reagents were then utilized to evaluate nephron segment development. Embryos were collected from pairwise matings of *ppargc1a$^{sa13186+/-}$* adult carriers and fixed at the 28 ss. For the knockdowns, WT embryos were microinjected at the one-cell stage with either MO1, SB MO1 or SB MO2 and similarly fixed at the 28 ss. WISH was completed on the *ppargc1a*-deficient embryo cohorts using segment-specific riboprobes to assess formation of the PCT, PST, DE and DL. Both *ppargc1a$^{sa13186-/-}$*mutants and knockdown embryos had a significantly expanded PST segment and a significantly decreased DL segment (*Figure 2B–D*, *Figure 2—figure supplements 3* and *4*). In contrast, there were no significant changes in PCT or DE segment formation (*Figure 2—figure supplement 2A–C*).

To further verify the specificity of the phenotypes, we performed rescue studies in *ppargc1a* mutant embryos. Full-length *ppargc1a* capped mRNA (cRNA) was synthesized in vitro, purified and microinjected at the one-cell stage into clutches obtained from pairwise matings of *ppargc1a$^{sa13186+/-}$* adults. Segmentation was assessed by WISH at the 28 ss to evaluate the development of the PST and DL segments. *ppargc1a* cRNA was sufficient to rescue PST and DL segment length in *ppargc1a$^{sa13186-/-}$* mutant embryos (*Figure 2B–D*). This result confirmed that the nephron

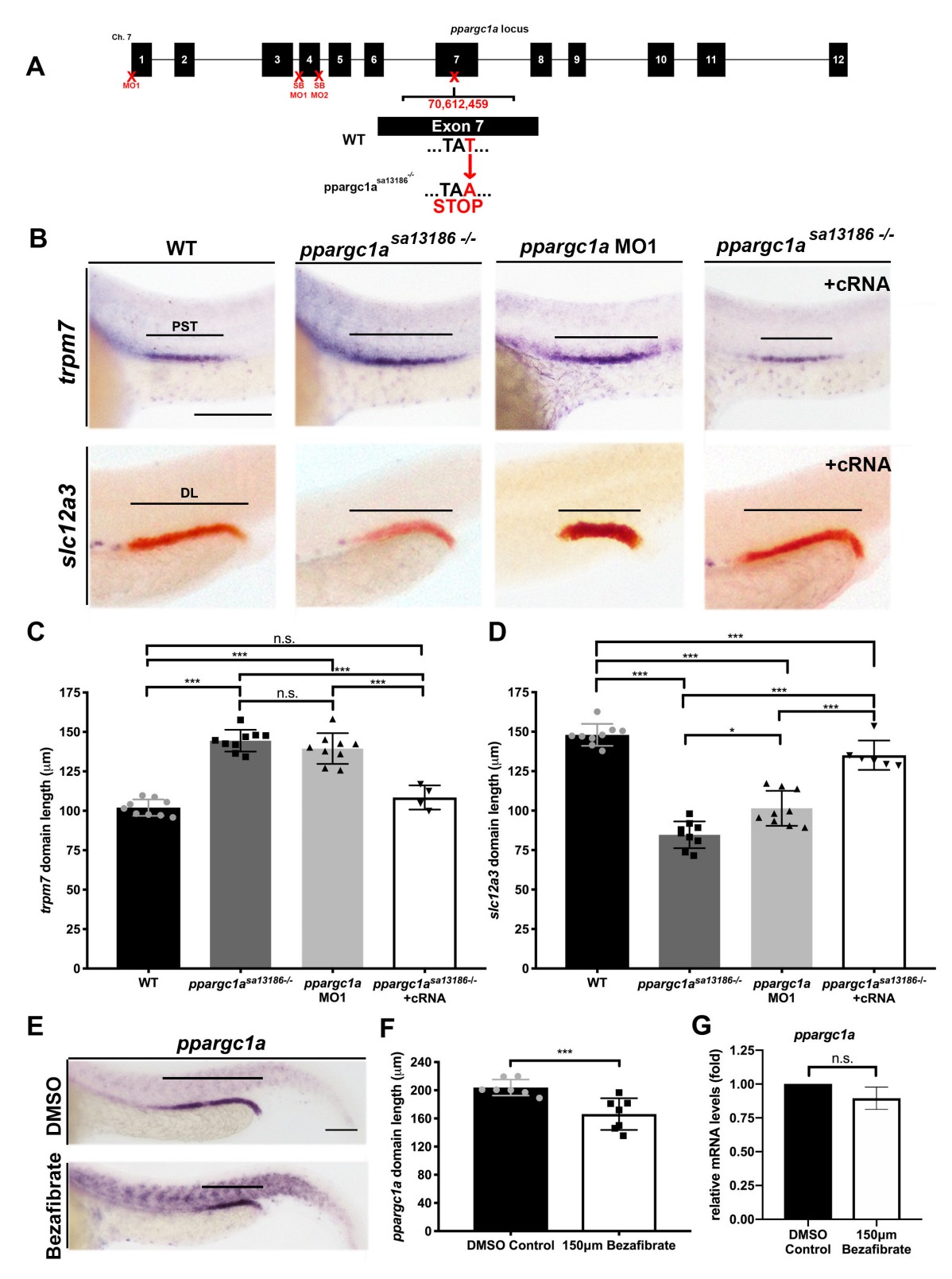

**Figure 2.** Loss-of-function studies show *ppargc1a* is necessary for proper PST and DL formation. (**A**) Exon map of zebrafish *ppargc1a* and the target sites (X) for morpholinos (MO) and the location of the *ppargc1a*^sa13186^ mutant allele. (**B**) WISH images of 28 ss *ppargc1a*^sa13186^ WT siblings (WT), *ppargc1a*^sa13186-/-^, *ppargc1a* MO1 injected, and *ppargc1a*^sa13186-/-^ + *ppargc1a* cRNA illustrating the changes in PST (*trpm7*-purple, top) and DL (*slc12a3*-red, bottom) formation in the *ppargc1a*^sa13186-/-^ and *ppargc1a* morphants, and the subsequent rescue when *ppargc1a* cRNA was added to the
*Figure 2 continued on next page*

Figure 2 continued

*ppargc1a*$^{sa13186-/-}$. Scale bars = 100 μm. Absolute length measurements of the PST (C), and DL (D) segments. (E) *ppargc1a* expression in DMSO control (top) and 150 μM bezafibrate-treated (bottom) zebrafish at the 28 ss following vehicle or vehicle/drug addition at the 5 hpf stage. Scale bars = 65 μm. (F) Absolute length measurements of the *ppargc1a* expression domain at 28 ss in DMSO control and bezafibrate-treated embryos from panel E. (G) qRT-PCR results showing *ppargc1a* RNA expression levels in bezafibrate-treated samples compared to DMSO controls. Data are represented as ±SD, significant by t test, n.s. = not significant, ** = p < 0.01, *** = p < 0.001.

DOI: https://doi.org/10.7554/eLife.40266.005

The following figure supplements are available for figure 2:

**Figure supplement 1.** *ppargc1a* is conserved across vertebrate species.

DOI: https://doi.org/10.7554/eLife.40266.006

**Figure supplement 2.** *ppargc1a* loss of function does not affect PCT or DE segment development.

DOI: https://doi.org/10.7554/eLife.40266.007

**Figure supplement 3.** Loss of function via a *ppargc1a* splice blocking MO recapitulates the decreased DL phenotype seen in other loss of function tests.

DOI: https://doi.org/10.7554/eLife.40266.008

**Figure supplement 4.** Loss of function via a *ppargc1a* splice blockingMO recapitulates the decreased DL phenotype seen in other loss of function tests.

DOI: https://doi.org/10.7554/eLife.40266.009

**Figure supplement 5.** The time of bezafibrate addition causes differential effects on *ppargc1a* domain length, and the effect of bezafibrate on DL development is not rescued by *ppargc1a* RNA overexpression.

DOI: https://doi.org/10.7554/eLife.40266.010

**Figure supplement 6.** Pronephros and body length measurements indicate no significant change in any treatment group compared to WT.

DOI: https://doi.org/10.7554/eLife.40266.011

phenotypes observed in this mutant model are caused by specific disruption of *ppargc1a* and exclude the possibility of other underlying genetic alterations.

Next, we explored whether there was a connection between the outcomes of bezafibrate treatment and *ppargc1a* loss of function during nephron segmentation. Previous publications have reported that PPAR agonists, including bezafibrate, can cause an increase or decrease of *ppargc1a/PGC1a* expression in cells and tissues in a context-dependent manner (*Pardo et al., 2011*; *Liao et al., 2010*; *Sanoudou et al., 2010*; *Goto et al., 2017*; *Wang and Moraes, 2011*). Since bezafibrate treatment and *ppargc1a* deficiency caused matching segment phenotypes, we hypothesized that bezafibrate decreased *ppargc1a* expression in renal progenitors. To test this, WT embryos were treated with either vehicle control or 150 μM bezafibrate/DMSO beginning at different developmental times (4 hpf, 5 hpf, 6 hpf, 8 hpf, 9 hpf, 10 hpf, 5 ss and 10 ss), incubated until the 28 ss, and then WISH was performed. Compared to WT controls, bezafibrate treatment resulted in a significant decrease in the expression domain of *ppargc1a* in the pronephros when the drug was added between 4 and 9 hpf (*Figure 2E,F*, *Figure 2—figure supplement 5A,B*). Next, we explored whether the *ppargc1a* expression domain was altered at the 15 ss when it is normally expressed in the *pax2a*$^+$ renal progenitor domain. Embryos were treated with DMSO vehicle or 150 μM bezafibrate/DMSO at the 5 hpf stage, and then fixed at 15 ss for WISH. Interestingly, the *ppargc1a* expression domain in renal progenitors at 15 ss was not altered by the bezafibrate treatment (*Figure 2—figure supplement 5C*). We also performed qRT-PCR on pools of 28 ss embryos treated with DMSO vehicle or 150 μM bezafibrate/DMSO at the 5 hpf stage to quantify *ppargc1a* expression levels. There was no significant difference in total *ppargc1a* mRNA levels between WT and bezafibrate-treated embryos (*Figure 2G*). Furthermore, we found that *ppargc1a* RNA overexpression was not sufficient to rescue DL development at the 28 ss in embryos treated with bezafibrate at the 5 hpf stage (*Figure 2—figure supplement 5D,E*). Additionally, we assessed the overall morphology as well as pronephros formation in 28 ss embryos treated with bezafibrate or vehicle control at the 5 hpf, as well as wild-type and *ppargc1a*-deficient embryos (*Figure 2—figure supplement 6*). Analysis of body length and pronephros length, the latter through WISH to detect expression of the pan-tubule and duct marker *cdh17*, showed no statistically significant differences between the groups (*Figure 2—figure supplement 6*). Taken together, these results are consistent with the notion that the *ppargc1a* expression domain is reduced in bezafibrate-treated embryos because the DL is reduced, and not specifically due to the loss of *ppargc1a* activity.

## Loss of *ppargc1a* does not change cellular turnover in the developing nephrons

Segmentation of the renal progenitors in the intermediate mesoderm occurs from the early somito-genesis stages through to the 28 ss based on the detection of molecularly distinct regions that emerge and then show dynamic alterations over this developmental time period, all while the renal progenitors are also undergoing a mesenchymal to epithelial transition (*Wingert et al., 2007*; *Wingert and Davidson, 2011*; *Li et al., 2014*; *Gerlach and Wingert, 2014*; *McKee et al., 2014*; *Kroeger and Wingert, 2014*; *Cheng and Wingert, 2015*; *Marra and Wingert, 2016*; *Drummond et al., 2017*; *Poureetezadi et al., 2016*). The proliferation and caudal migration of renal precursors has also been reported to impact pronephros segment size (*Naylor et al., 2016*). To this end, we wanted to determine when the loss of *ppargc1a* first presented significant changes to the emerging segment domains, and to address if these changes were coincident with alterations in cellular dynamics in the nephron field.

A series of WISH studies were performed with PST and DL markers to compare these emerging segment populations in WT controls and *ppargc1a*-deficient embryos (*Figure 3*). We found that the earliest time point of divergence between WT and *ppargc1a*-deficient embryos occurred at the 20 ss, when there was a distinction in the expression domain of both the PST marker *trpm7* and DL marker *slc12a3* (*Figure 3A–D*, *Figure 3—figure supplement 1*). *ppargc1a*-deficient embryos displayed a significant increase in the emerging PST length and a significant decrease in the emerging DL length (*Figure 3A–D*). These changes correlate with the result that *ppargc1a* mutants and morphants exhibit a longer PST and shortened DL when segmentation is completed (*Figure 2*).

At this pivotal 20 ss time point, we then sought to identify whether either of these changes were associated with regional fluctuations in cell birth or death. To assess this, we combined FISH with whole mount immunohistochemistry and confocal imaging to assess the PST and DL. WT control and *ppargc1a*-deficient embryos were fixed at the 20 ss and nephron cells were detected based on *trpm7* (PST) or *slc12a3* (DL) transcripts in combination with either anti-Caspase-3 to detect cell death or anti-phospho-Histone H3 (anti-pH3) to label proliferating cells (*Kroeger et al., 2017*). The results showed that there was no significant difference between WT and *ppargc1a*-deficient embryos in the number of $trpm7^+$/anti-Caspase-$3^+$ cells (*Figure 3E,I*, *Figure 3—figure supplement 2*) or $slc12a3^+$/anti-Caspase-$3^+$ cells (*Figure 3G,J*, *Figure 3—figure supplement 4*). Quantification of $trpm7^+$/pH3$^+$ cell number (*Figure 3F,I*, *Figure 3—figure supplement 3*) and $slc12a3^+$/pH3$^+$ cell number (*Figure 3H,J*, *Figure 3—figure supplement 5*) also showed that there was no statistically significant difference between WT and *ppargc1a*-deficient groups. The results from these experiments suggest that there are no significant changes in cellular turnover driving the PST and DL segment boundary changes that occur in *ppargc1a*-deficient embryos.

## *ppargc1a* promotes DL segment formation by positively regulating the expression domain of the *tbx2b* transcription factor in nephron precursors

To gain insight into how *ppargc1a* influences nephron segmentation, we next explored its relationship with the T-box transcription factor *tbx2b*, which was recently shown to be essential for DL formation (*Drummond et al., 2017*). Transcripts encoding *tbx2b* are highly expressed in the distal regions of the zebrafish pronephros, and loss of *tbx2b* results in a significantly decreased DL segment size (*Drummond et al., 2017*). To test the relationship between *ppargc1a* and *tbx2b*, WISH was performed on *ppargc1a* deficient embryos to assess *tbx2b* expression. Compared to WT controls, the *tbx2b* expression domain in the nephron was significantly reduced in $ppargc1a^{sa13186-/-}$ mutants (*Figure 4A,B*) as well as *ppargc1a* morphants (data not shown). These results led us to hypothesize that loss of *tbx2b* expression underlies the decreased DL segment domain when Ppargc1a activity is compromised. When the reciprocal experiment was performed in *tbx2b*-deficient embryos (*Drummond et al., 2017*; *Gross and Dowling, 2005*), we detected no change in *ppargc1a* expression (*Figure 4C,D*), consistent with the notion that *ppargc1a* is upstream of *tbx2b*.

To test this further, we examined whether overexpression of *tbx2b* was sufficient to rescue DL development in *ppargc1a* mutants. *tbx2b* cRNA was injected at the one-cell stage into clutches obtained from matings of $ppargc1a^{sa13186+/-}$ adults, and segmentation was assessed at the 28 ss by WISH using our DL-specific riboprobe followed by genotype analysis (*Figure 4E*). While

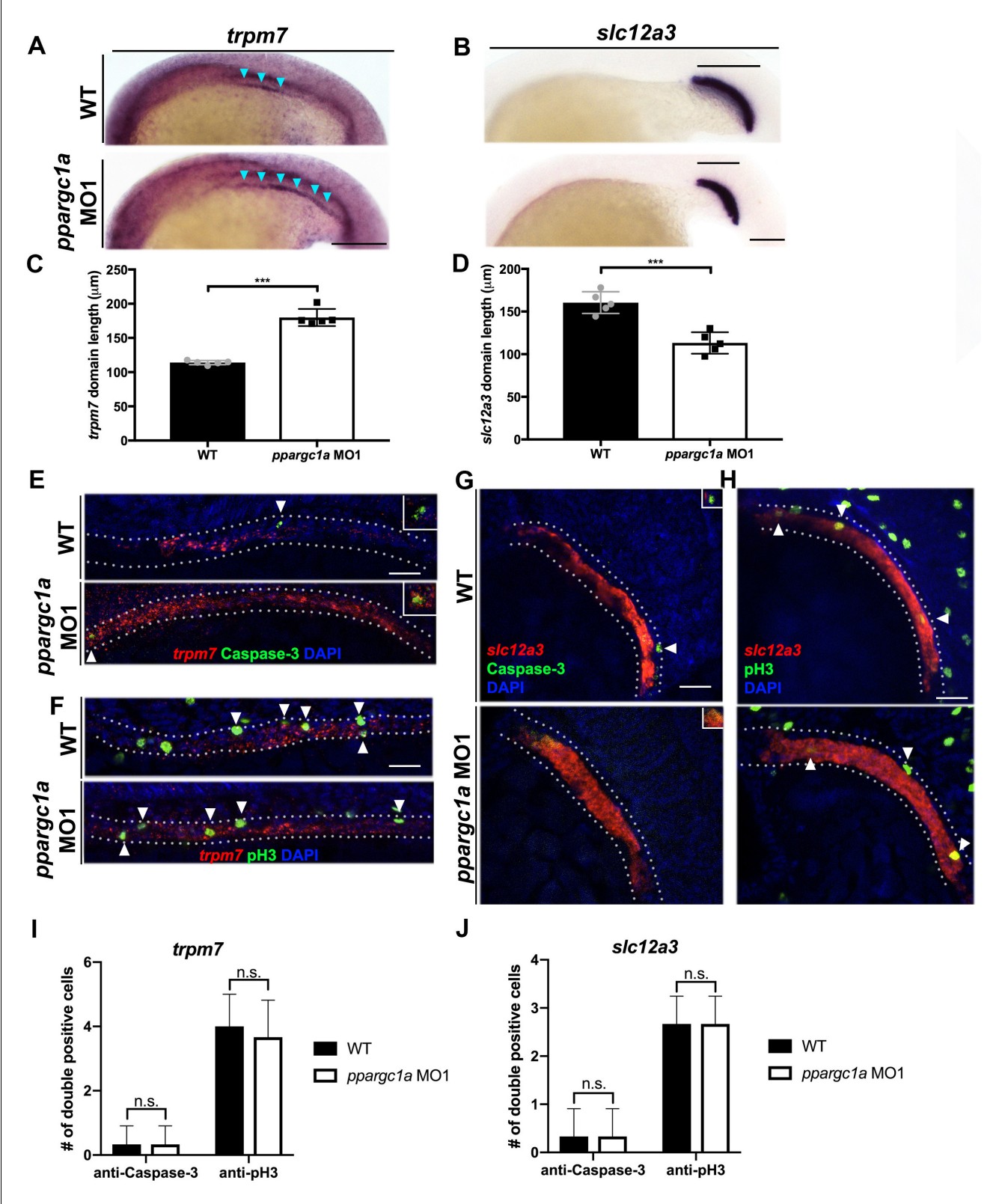

**Figure 3.** Loss of *ppargc1a* does result in segment changes at the 20 ss but no change in cellular turnover is observed. (**A**) WISH at the 20 ss for *trpm7* (left) and (**B**) *slc12a3* (right) in WT (top) and *ppargc1a* MO (bottom). Blue arrowheads indicate *trpm7* expression in the developing pronephros. Scale bar = 100 μm. The representative graphs showing absolute length measurements of *trpm7* (**C**) and *slc12a3* (**D**). (**E**) FISH/IHC for *trpm7* (red) and anti-Caspase-3 (green) with DAPI (blue) in WT (top) and *ppargc1a* MO1 (bottom). (**F**) FISH/IHC for *trpm7* (red) and anti-phospho-Histone H3 (green) in WT
*Figure 3 continued on next page*

*Figure 3 continued*

(top) and *ppargc1a* MO1 (bottom). (**G**) FISH/IHC for *slc12a3* (red) and anti-Caspase-3 (green) with DAPI (blue) in WT (top) and *ppargc1a* MO1 (bottom). (**H**) FISH/IHC for *slc12a3* (red) and anti-phospho-Histone H3 (green) with DAPI (blue) in WT (top) and *ppargc1a* MO1 (bottom). (**I**) The number of *trpm7*/Caspase-3 or *trpm7*/pH3 double-positive cells is depicted. (**J**) The number of *slc12a3*/Caspase-3 or *slc12a3*/pH3 double positive cells is depicted. (WT quantifications = black bars, *ppargc1a*-deficient quantifications = white bars.) Data are represented as ±SD, significant by t test, n.s. = not significant, \*\*\* = $p < 0.001$.

DOI: https://doi.org/10.7554/eLife.40266.012

The following figure supplements are available for figure 3:

**Figure supplement 1.** Decreased DL phenotypes are evident at the 25 ss and 27 ss in *ppargc1a*-deficient embryos.
DOI: https://doi.org/10.7554/eLife.40266.013
**Figure supplement 2.** Confocal image split channels of *trpm7* and anti-Caspase-3 in WT and *ppargc1a* deficient 20 ss zebrafish.
DOI: https://doi.org/10.7554/eLife.40266.014
**Figure supplement 3.** Confocal image split channels of *trpm7* and anti-pH3 in WT and *ppargc1a* deficient 20 ss zebrafish.
DOI: https://doi.org/10.7554/eLife.40266.015
**Figure supplement 4.** Confocal image split channels of *slc12a3* and anti-Caspase-3 in WT and *ppargc1a*-deficient 20 ss zebrafish.
DOI: https://doi.org/10.7554/eLife.40266.016
**Figure supplement 5.** Confocal image split channels of *slc12a3* and anti-pH3 in WT and *ppargc1a*-deficient 20 ss zebrafish.
DOI: https://doi.org/10.7554/eLife.40266.017

*ppargc1a*[sa13186-/-] mutants displayed the hallmark short DL segment, there was no significant difference in DL length between WT controls and *ppargc1a*[sa13186-/-] mutants that received *tbx2b* cRNA, indicating that *tbx2b* provision had rescued DL segment development (*Figure 4E,F*). Taken together, these results indicate that Ppargc1a regulates *tbx2b*, either directly or indirectly, to control formation of the DL segment.

### *ppargc1a* regulates PST boundary formation through a reciprocally antagonistic relationship with the *sim1a* transcription factor

During zebrafish embryonic nephron segmentation, *sim1a* is necessary and sufficient for formation of the PST segment as well as the Corpuscle of Stannius (CS), the latter being an endocrine gland in teleost fish which arises from the intermediate mesoderm, where CS precursors are intermingled with distal segment precursors (*Cheng et al., 2015*). Thus, we sought to delineate the relationship between *ppargc1a* and *sim1a*. To do this, we examined *ppargc1a* expression in *sim1a*-deficient embryos in which transcript splicing is abrogated through morpholino knockdown (*Figure 5—figure supplement 1*) (*Löhr et al., 2009*; *Cheng and Wingert, 2015*). Interestingly, we found that the domain of *ppargc1a* expression in renal progenitors was significantly increased in length in the *sim1a*-deficient embryos compared to WT controls (*Figure 5A,D*). This result suggested *sim1a* was possibly upstream of *ppargc1a*.

Since *ppargc1a*[sa13186-/-] mutants evince an increased PST segment (*Figure 2B,C*) in addition to an increased CS size (*Figure 5—figure supplement 2A,B*), we next examined whether Ppargc1a deficiency was associated with changes in *sim1a* expression in renal precursors. WISH was performed to investigate the pattern of *sim1a* expression in *ppargc1a*[sa13186-/-] mutant embryos. This analysis revealed that *ppargc1a*[sa13186-/-] mutant embryos had an increased *sim1a* domain at the 20 ss, a time point that coincides with *sim1a* expression in the PCT and PST segments, as well as at the 28 ss, which coincides with expression of *sim1a* in the CS anlage (*Figure 5B,E*, *Figure 5—figure supplement 2C,D*). Taken together, these results suggested that there were reciprocal antagonistic interactions between these two factors, which act to delineate segmental domains.

To further explore cross-repressive interactions, we next performed RNA overexpression studies with *sim1a* and *ppargc1a* in WT embryos. Interestingly, overexpression of *sim1a* cRNA led to a statistically significant decrease in the domain of *ppargc1a* transcript expression within the pronephros at the 28 ss (*Figure 5A,D*), as well as an increase in PST segment length (*Figure 5C,F*), the latter as previously reported (*Cheng and Wingert, 2015*). *sim1a* cRNA overexpression also had no effect on DL segment development (*Figure 5—figure supplement 3*), consistent with intact *ppargc1a* expression in the distal pronephros region (*Figure 5A*). Overexpression of *ppargc1a* cRNA caused a statistically significant decrease in the domain of *sim1a* transcript expression within the pronephros at the 20 ss (*Figure 5B,E*) as well as the CS anlage at the 28 ss (*Figure 5—figure supplement 2C,D*).

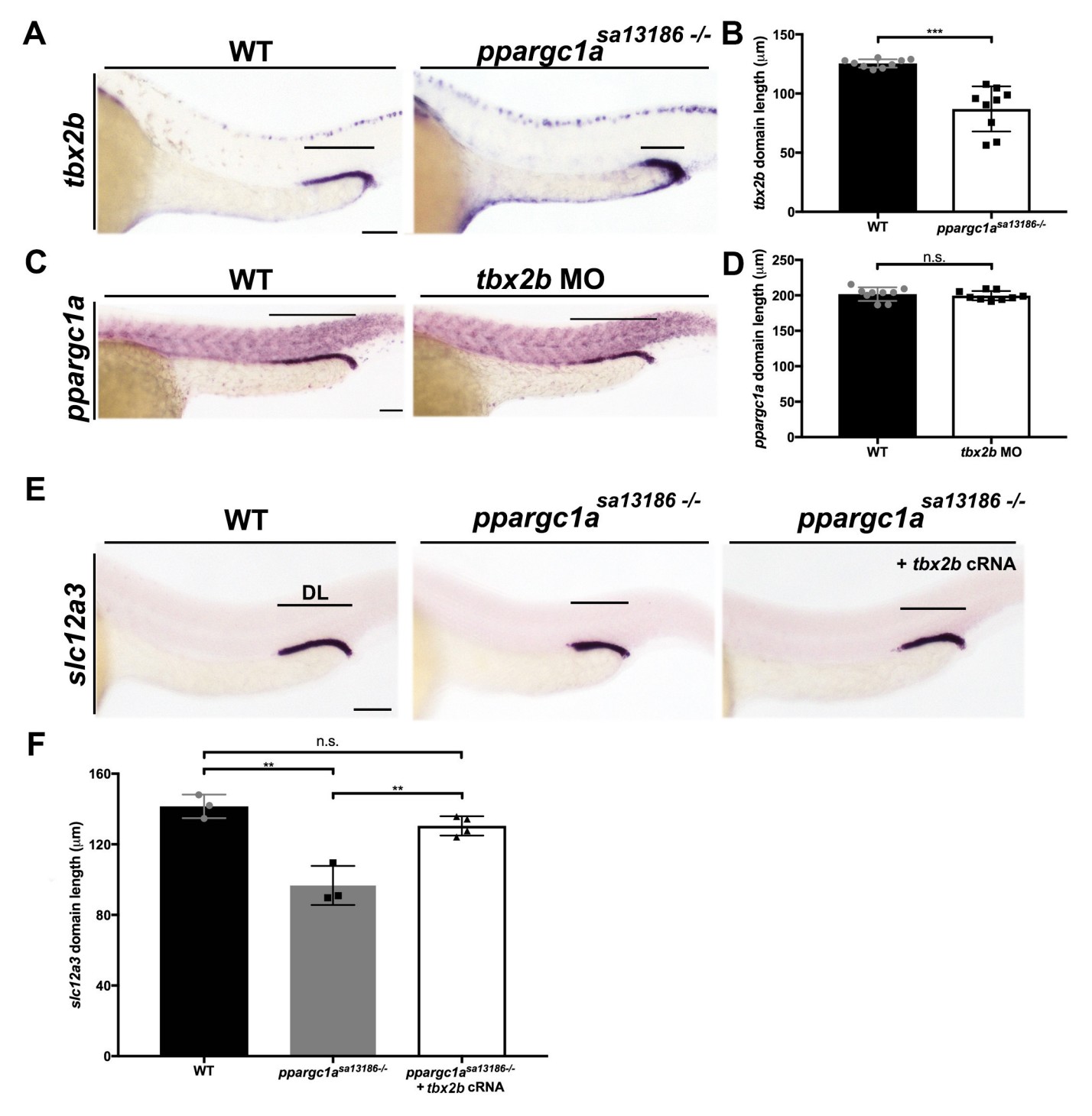

**Figure 4.** *ppargc1a* acts upstream of *tbx2b* to form the DL segment. (A) WISH for *tbx2b* (purple) expression in 28 ss WT and *ppargc1a*[sa13186-/-] zebrafish. Scale bars = 75 µm. (B) Absolute length measurements of *tbx2b* mRNA expression domains in WT and *ppargc1a*[sa13186-/-] zebrafish. (C) *ppargc1a* expression in WT and *tbx2b* MO injected 28 ss zebrafish. Scale bars = 65 µm. (D) Absolute length measurements of *ppargc1a* expression domain in WT and *tbx2b* MO injected zebrafish. (E) WISH at 28 ss for *slc12a3* (purple) in WT, *ppargc1a*[sa13186-/-], and *ppargc1a*[sa13186-/-] injected with *tbx2b* cRNA. Scale bars = 75 µm. (F) Absolute length measurements of *slc12a3* mRNA expression domains in WT, *ppargc1a*[sa13186-/-], and *ppargc1a*[sa13186-/-] + *tbx2b* cRNA. Data are represented as ±SD, significant by t test, n.s. = not significant, ** = p < 0.01, and *** = p < 0.001.
DOI: https://doi.org/10.7554/eLife.40266.018

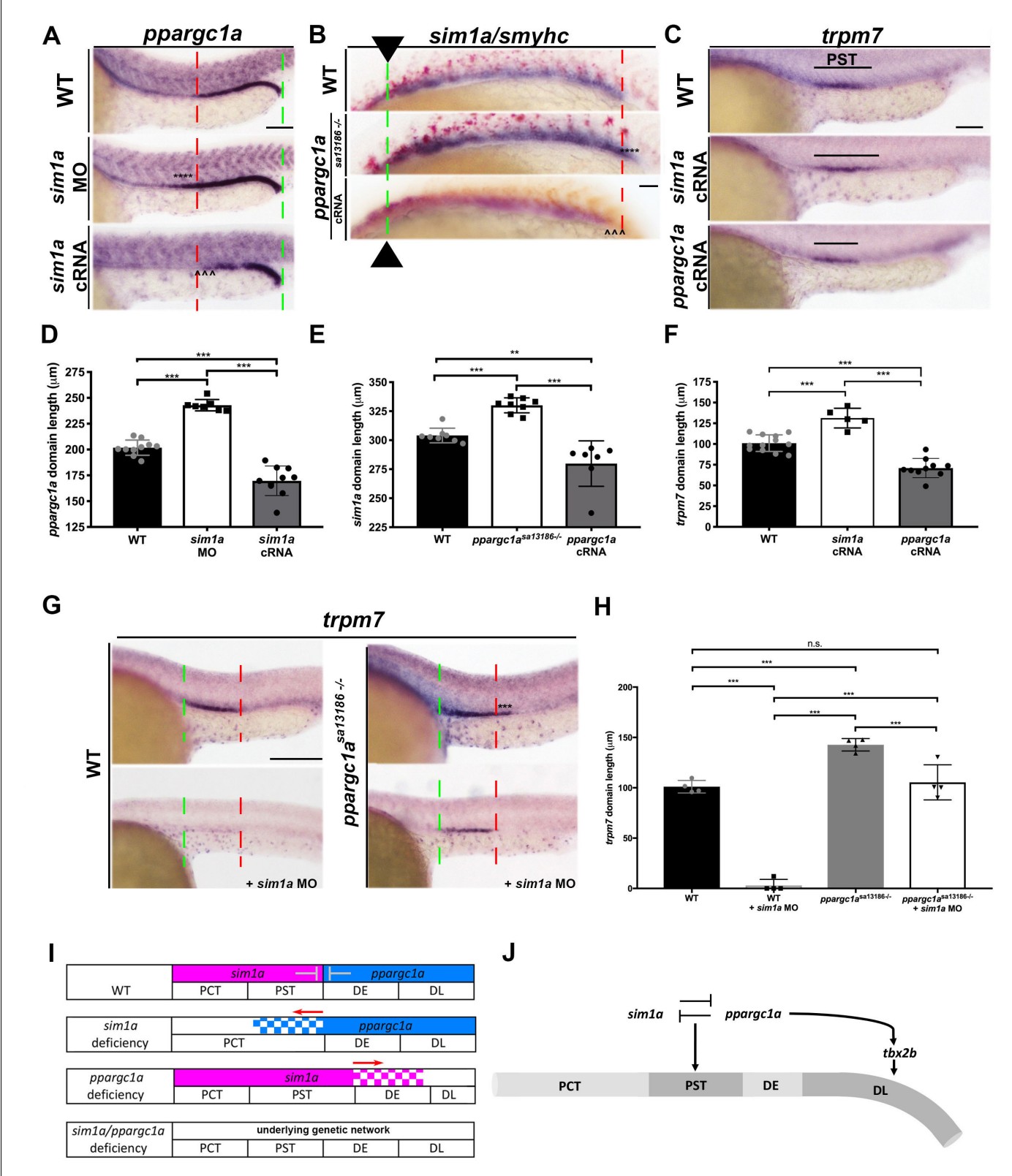

**Figure 5.** *ppargc1a* and *sim1a* have a reciprocal antagonistic relationship that is necessary to negotiate the PST segment boundary. (**A**) WISH for *ppargc1a* (purple) expression at 28 ss in WT, *sim1a* MO, and *sim1a* cRNA injected zebrafish. The green and red dashed lines indicate the beginning and the end of WT expression, respectively, the expansion of the *ppargc1a* domain in *sim1a* MO is denoted with asterisks, and the reduction in the *ppargc1a* domain in *sim1a* cRNA injected is denoted with arrowheads. Scale bars = 60 μm. (**B**) WISH for *sim1a* (purple) and *smyhc* (red) in 20 ss WT, *Figure 5 continued on next page*

*Figure 5 continued*

*ppargc1a^{sa13186-/-}*, and *ppargc1a* cRNA injected zebrafish. Arrows and the dashed green line represent the beginning of the *sim1a* domain expression, the red dashed line indicate the end of the WT *sim1a* expression domain to illustrate the expansion of the segment boundary in *ppargc1a^{sa13186-/-}* is denoted with asterisks, and the reduction of the *sim1a* domain in *ppargc1a* cRNA injected is denoted with arrowheads. Scale bar = 40 μm. (C) WISH for *trpm7* at 28 ss in WT, *sim1a* cRNA, and *ppargc1a* cRNA injected zebrafish. Scale bar = 50 μm (D) Absolute length measurements of the *ppargc1a* domain in 28 ss in WT, *sim1a* MO injected, and *sim1a* cRNA injected zebrafish. (E) Absolute length measurements of the 20 ss *sim1a* domain in WT, *ppargc1a^{sa13186-/-}*, and *ppargc1a* cRNA injected zebrafish. (F) Absolute length measurements of the *trpm7* domain in the 28 ss WT, *sim1a* cRNA injected, and *ppargc1a* cRNA injected zebrafish. (G) WISH for *trpm7* (PST segment) in WT, WT + *sim1a* MO, *ppargc1a^{sa13186-/-}*, and *ppargc1a^{sa13186-/-}* + *sim1a* MO zebrafish. The green and red dashed lines indicate the beginning and end of WT *trpm7* expression domains respectively. Asterisks represent expanded expression domain. Scale bars = 100 μm. (H) Absolute length measurements for the *trpm7* domain in WT, *sim1a* MO injected, *ppargc1a^{sa13186-/-}*, and *ppargc1a^{sa13186-/-}* *sim1a* MO injected zebrafish. (I) Expression summary table depicting the segment boundaries and expression domains of *ppargc1a* and *sim1a* in WT, *sim1a*-deficient, *ppargc1a*-deficient, and double-deficient zebrafish. (J) Genetic model illustrating the relationships supported in this study that *ppargc1a* works upstream of *tbx2b* to form the DL and has a cross-repressive relationship with *sim1a* to properly form the segment boundary of the PST. Data are represented as ±SD, significant by t test, n.s. = not significant, *** = p < 0.001.
DOI: https://doi.org/10.7554/eLife.40266.019

The following figure supplements are available for figure 5:

**Figure supplement 1.** *sim1a* knockdown was verified by reverse transcriptase PCR and known *sim1a* deficient phenotypes.
DOI: https://doi.org/10.7554/eLife.40266.020

**Figure supplement 2.** The CS is expanded in *ppargc1a*-deficient zebrafish.
DOI: https://doi.org/10.7554/eLife.40266.021

**Figure supplement 3.** *sim1a* overexpression does not affect DL segment development.
DOI: https://doi.org/10.7554/eLife.40266.022

*ppargc1a* cRNA overexpression also reduced the PST segment length at the 28 ss (*Figure 5C,F*). These results support the conclusion that *sim1a* and *ppargc1a* have repressive effects on each other in the context of renal progenitor development in the zebrafish.

In light of the observation that *sim1a* cRNA overexpression is sufficient to expand the PST segment (*Figure 5C,E*) (*Cheng and Wingert, 2015*), we hypothesized that the expansion of the *sim1a* expression in *ppargc1a* mutants was responsible for the enlarged PST phenotype. To interrogate this and gain further insight about the relationship between *ppargc1a* and *sim1a*, we tested the outcome of *sim1a* knockdown in *ppargc1a* mutants compared to WT embryos. *sim1a* MO was injected at the one-cell stage into *ppargc1a^{sa13186}* heterozygous in-crosses, and nephron segmentation was assessed in these and uninjected controls by WISH at the 28 ss to study PST development. Consistent with our previous results, *sim1a* deficiency abrogated the PST in WT embryos, and there was a larger PST in the *ppargc1a^{sa13186-/-}* mutants (*Figure 5G,H*). By comparison, *ppargc1a/sim1a* doubly deficient embryos formed a PST segment of normal length (*Figure 5G,H*). Taken together, these results suggest that *sim1a* promotes PST fate by acting to control the expression domain of *ppargc1a* in proximal renal progenitors, which establishes the proper boundary of the PST segments (*Figure 5I*). Additionally, these results show that *ppargc1a* serves an antagonistic role to *sim1a*, restricting the spatial domain of *sim1a* expression from distal renal progenitors to define the PST segment boundary (*Figure 5I*).

## Discussion

Elucidating the genetic regulators that direct cell fate decisions during nephron ontogeny is paramount to understanding how molecular changes cause renal organogenesis defects, and can be applied to advance regenerative medicine approaches for the treatment of kidney disease. Evidence from our chemical genetic screen led us to identify that PPAR signaling was a possible candidate for regulating the nephron segment lineages during embryogenesis. Here, we determined that *ppargc1a* expression in renal progenitors is essential to mitigate segment fate choices that establish segment identities in the pronephros.

Specifically, we discovered that *ppargc1a* is necessary for proper formation of the PST and DL segment boundaries. We also identified the timing associated with these segmentation changes and observed cellular turnover analogous to WT embryos, suggesting that segment phenotypes in *ppargc1a* mutants are not related to alterations in cell proliferation or cell death. We ascertained that there are two critical genetic pathways that *ppargc1a* regulates to control PST and DL segment

size. Through a series of genetic studies, we determined that *ppargc1a* acts upstream of *tbx2b* to promote DL formation and we uncovered an intriguing, reciprocally antagonistic relationship between *sim1a* and *ppargc1a* that operates to properly form the PST segment (*Figure 5J*), as well as the CS anlage (*Figure 5—figure supplement 2*). Discovery of the opposing activities of *ppargc1a* and *sim1a* in the present work highlights for the first time how the precise dimension of the PST is defined by reciprocal antagonism during segmentation of the kidney nephron unit. While this interplay is essential for segment fate choice, a fascinating aspect revealed by our studies is that there is an underlying genetic network that enables PST segment development to transpire normally in the absence of both of these powerful opposing transcription factors. Future efforts to identify these other genetic components are needed to decipher the mechanisms of PST formation, where there have been few advances in understanding the patterning of this segment despite progress in understanding ciliated cell fate choice in this pronephros region (*Marra et al., 2016*). With respect to DL segment ontogeny, additional work is also needed to ascertain how *ppargc1a* relates to known DL regulators such as the transcription factors *mecom*, *tbx2a*, and *emx1* along with prostaglandin signaling (*Li et al., 2014*; *Drummond et al., 2017*; *Poureetezadi et al., 2016*; *Morales et al., 2018*).

Previous studies have established that PGC-1α can exert its regulatory effects on the transcription of target genes in cell-specific contexts through its interactions with a variety of nuclear receptors. Transcriptional regulation by PGC-1α is known to play key roles in diverse biological processes, from mitochondrial biogenesis to metabolic activities, in which PGC-1α coordinates dynamic responses to physiological demands (*Lin et al., 2005*). Additionally, PGC-1α coordinates transcriptional activities during cell differentiation, such as in erythrocyte maturation, where it has unique and shared nuclear targets with its family member PGC-1α that impact globin gene regulation (*Cui et al., 2014*). Transcriptional profiling of renal progenitors in *ppargc1a* mutants, assessment of chromatin state with techniques such as the assay for transposase accessible chromatin using sequencing (ATAC-seq) and chromatin immunoprecipitation with sequencing (ChIP-Seq), can delineate the possible direct and indirect targets of Ppargc1a. As coactivators typically function in multiprotein complexes, identification of both the relevant nuclear receptor targets and other binding partners of Ppargc1a in renal progenitors will be crucial to gaining additional insight on the emergence of nephron segment populations and the establishment of boundaries between adjacent segments. The precedence that members of the PGC-1 family can have redundant activities suggests that future investigations should also explore this possibility for Ppargc1a during nephron segmentation (*Cui et al., 2014*). Such redundancy might explain the absence of congenital kidney defects in the PGC-1α murine knockout, and may necessitate combined deficiency studies to delineate the roles of PGC-1 members in mammalian renal development.

Further explorations of the mechanisms by which Ppargc1a regulates renal progenitors should also consider possibilities in addition to transcriptional control because of the precedence that PGC-1α possesses a number of molecular activities depending on the context. PGC-1α is highly versatile, whereby it can interact with a range of molecules other than transcription factors. PGC-1α can recruit histone acetyl transferase containing coactivator proteins and can also interact with RNA processing complexes (*Knutti and Kralli, 2001*; *Puigserver and Spiegelman, 2003*). Proteomics approaches in renal progenitors may thus identify Ppargc1a binding partners with such activities and if so, will highlight other directions by which to elucidate the roles of Ppargc1a during nephron segmentation.

While functions for Ppargc1a in vertebrate renal development have not been reported until the present study, interesting roles of PGC-1α have been identified during the response to kidney damage in mammals. PGC-1α serves a renoprotective function in the murine kidney. PGC-1α is downregulated after ischemic and sepsis induced acute kidney injury, and normal renal function is restored with overexpression of PGC-1α (*Tran et al., 2011*; *Tran et al., 2016*). PGC-1α is also reduced in three different modes of chronic kidney disease: toxic, obstructive, and genetic (*Han et al., 2017*). *Hes1* represses *ppargc1a* during Notch-induced renal fibrosis; however, induced overexpression of *ppargc1a* can ameliorate this process (*Han et al., 2017*). In light of the potent influence that Ppargc1a exacts on renal progenitors during embryonic kidney development, it is intriguing to speculate whether modulation of Ppargc1a could be utilized further to stimulate regenerative therapies. In addition, Ppargc1a is a prime candidate for being involved in the capacity of the zebrafish adult to regenerate nephrons and undergo neonephrogenesis, where the prediction would be that Ppargc1a is similarly essential in nephron precursors to mitigate emergence of the PST and DL

segment fates (*Diep et al., 2011*; *McCampbell and Wingert, 2014*; *McCampbell et al., 2014*; *McCampbell et al., 2015*; *Drummond and Wingert, 2016*).

While there have been ongoing advancements in our understanding of nephron patterning during development and the pathways that facilitate nephron epithelial regeneration following damage, many gaps in knowledge still remain. The continued identification of the genetic networks that regulate renal progenitors in these contexts has far-reaching implications (*Chambers and Wingert, 2016*). Our new insights into nephron segmentation have divulged novel mechanisms that define nephron segment boundaries in the embryonic renal mesoderm. Taken together, our data show for the first time that *ppargc1a* is required for pivotal renal progenitor fate decisions that establish nephron segment pattern during kidney development. Given the fundamental conservation of segment pattern across vertebrate nephrons, we speculate that these newly discovered roles of Ppargc1a will provide useful clues about PGC-1α functions in mammalian kidney development. As alterations in Ppargc1a activity have potent effects on nephron segment fate, our results suggest that Ppargc1a/PGC-1α may be an important molecular target for medical applications or engineering approaches involving the directed differentiation of pluripotent stem cells to fashion kidney organoids (*Chambers et al., 2016*). Here, we have only begun to appreciate the importance of *ppargc1a* in kidney development, focusing on the role it plays in segment boundary formation.

# Materials and methods

## Key resources table

| Reagent type (species) or resource | Designation | Source or reference | Identifiers | Additional information |
|---|---|---|---|---|
| Antibody | anti-Caspase-3 (rabbit) | BD Biosciences | 559565 | dilution 1:100 |
| Antibody | phospho-Histone H3 (Ser10) (rabbit) | Millipore | 06–570 | dilution 1:200 |
| Antibody | Alexa Fluor anti-rabbit secondary (goat) | Invitrogen | A11037 | dilution 1:500 |
| Chemical compound, drug | dimethyl sulfoxide (DMSO) | American Bioanalytical | AB03091-00100 | |
| Chemical compound, drug | bezafibrate | Enzo Life Sciences | BML-GR211-0001 | |
| Genetic reagent (*Danio rerio*) | *ppargc1a*<sup>sa13186</sup> zebrafish line | Zebrafish International Resource Center (ZIRC) | ZMP:sa13186 | Zebrafish Mutation Project allele sa13186 |
| Commercial assay or kit | PCR purification kit | Qiagen | 28106 | |
| Commercial assay or kit | NdeI restriction endonuclease enzyme | New England BioLabs | R0111S | |
| Commercial assay or kit | TRIzol Reagent | Invitrogen | 15596018 | |
| Commercial assay or kit | qScript cDNA SuperMix | QuantaBio | VWR 101414–106 | |
| Commercial assay or kit | PerfeCTa SYBR Green SuperMix with ROX | QuantaBio | VWR 101414–160 | |
| Chemical compound, drug | mMESSAGE mMACHINE SP6 Transcription kit | Ambion | AM1340 | |
| Other | custom antisense morpholino oligonucleotide | Gene Tools, LLC | *ppargc1a* ATG MO1 (ZFIN: MO1-ppargc1a) | 5'–CCTGATTACACCT GTCCCACGCCAT–3' |
| Other | custom antisense morpholino oligonucleotide | Gene Tools, LLC | *ppargc1a* SB MO1 | 5'–GGAGCTTCTTCAG CTACAAACAGAG–3' |
| Other | custom antisense morpholino oligonucleotide | Gene Tools, LLC | *ppargc1a* SB MO2 | 5'–GGTGAGCAGCTA CCTTGGCAACAGC–3' |
| Other | custom antisense morpholino oligonucleotide | Gene Tools, LLC | *tbx2b* MO | 5'–CCTGTAAAAACTG GATCTCTCATCGG–3' |

*Continued on next page*

*Continued*

| Reagent type (species) or resource | Designation | Source or reference | Identifiers | Additional information |
|---|---|---|---|---|
| Other | custom antisense morpholino oligonucleotide | Gene Tools, LLC | *sim1a MO* | 5'–TGTGATTGTGTA CCTGAAGCAGATG–3' |
| Software, algorithm | Nikon Elements imaging software | Nikon | | |
| Software, algorithm | Graphpad Prism 8 | GraphPad Prism (https://www.graph pad.com/scientific-software/prism/) | | |
| Software, algorithm | ImageJ | ImageJ (https:// imagej.nih.gov/ij/) | | |

## Zebrafish husbandry and ethics statement

Zebrafish were maintained in the Center for Zebrafish Research at the University of Notre Dame. All studies were performed with approval of the University of Notre Dame Institutional Animal Care and Use Committee (IACUC), under protocol numbers 13–021 and 16–025. For experiments with WT zebrafish, we utilized the Tübingen strain. Embryos were raised and staged as described (*Kimmel et al., 1995*). For all molecular studies, embryos were incubated in E3 medium from fertilization through the desired developmental stage at 28°C, anesthetized with 0.02% tricaine, and then fixed for analysis using 4% paraformaldehyde/1 x phosphate buffered saline (*Westerfield, 1993*).

## Whole mount and fluorescent whole mount *in situ* hybridization (WISH, FISH)

WISH was performed as previously described (*Cheng et al., 2014*; *Galloway et al., 2008*; *Lengerke et al., 2011*) with antisense RNA probes either digoxigenin-labeled (*ppargc1a, slc20a1a, trpm7, slc12a1, slc12a3, sim1a, tbx2b, stc1, cdh17*) or fluorescein-labeled (*smyhc, slc12a3, -slc12a1, pax2a*) by *in vitro* transcription using IMAGE clone templates as previously described (*Wingert et al., 2007*; *O'Brien et al., 2011*; *Gerlach and Wingert, 2014*; *McKee et al., 2014*). FISH was performed as previously described (*Brend and Holley, 2009*; *Marra et al., 2017*). For all gene expression studies, every analysis was done in triplicate for each genetic model, in a blinded fashion, with sample sizes of at least n = 20 for each replicate. A minimum of 5 representative individuals from each replicate were imaged and quantified, then subjected to statistical analysis.

## Immunofluorescence (IF)

Whole mount IF experiments were completed as previously described (*Kroeger et al., 2017*; *Marra et al., 2017*). For cell death and proliferation assays, rabbit anti-Caspase-3 diluted 1:100 (BD Biosciences 559565) and rabbit phospho-Histone H3 (Ser10) antibody diluted 1:200 (Millipore 06–570) were used, respectively. Anti-rabbit secondary antibody (Alexa Fluor, Invitrogen) was diluted 1:500.

## Chemical treatments

Chemical treatments were completed as previously described (*Poureetezadi et al., 2014*; *Poureetezadi et al., 2016*). Bezafibrate (Enzo Life Sciences, BML-GR211-0001) was dissolved in 100% DMSO to make 1 M stocks and diluted to the working dosage. For segment analysis, treatments were completed in triplicate with sample sizes of at least 20 embryos per replicate.

## Genetic models

The *ppargc1a^sa13186^* line was obtained from ZIRC (Eugene, OR) (*Busch-Nentwich et al., 2013*). Mutant embryos and heterozygous adults were identified by performing PCR with the following primers flanking the mutation site: forward 5'–GGGCCGGCATGTGGAATGTAAAGACTTAAACA TGCCAACCTCCACTACTACGACATCATCGTTGTCTTCCACCCCCCCTTCGTCTTCCTCACTGGCCA GG–3' and reverse 5'–TCCCACTACCCCGCTATAGAAGGCTTGCTGAGGCTTTCCAAAGTGCTTG TTGAGCTCGTCCCGGATCTCCTGGTCCCTAAGAAGTTTCCTGCCACCAGAA–3'. PCR products

were purified (Qiagen) and sent to the Notre Dame Genomics Core for sequencing analysis or subjected to restriction enzyme digest with NdeI (New England BioLabs) and separation on a 2% agarose gel to identify WT, heterozygous, or mutant samples. Antisense morpholino oligonucleotides (MOs) were obtained from Gene Tools, LLC (Philomath, OR) and solubilized in DNase/RNase-free water to create 4 mM stock solutions which were then stored at −20°C. Zebrafish embryos were injected at the 1 cell stage with 1–2 nL of diluted MO. *ppargc1a* was targeted with the following: ATG MO1 'MO1' 5'–CCTGATTACACCTGTCCCACGCCAT–3' (400 μM) (*Hanai et al., 2007*; *Bertrand et al., 2007*), Splice MO1 'SB MO1' 5'–GGAGCTTCTTCAGCTACAAACAGAG–3' (400 μM), and Splice MO2 'SB MO2' 5'–GGTGAGCAGCTACCTTGGCAACAGC–3' (400 μM). *tbx2b* knockdowns were performed with an ATG MO 5'–CCTGTAAAAACTGGATCTCTCATCGG–3' (400 μM) (*Gross and Dowling, 2005*; *Drummond et al., 2017*). To target *sim1a*, a splice blocking MO 5'–TGTGATTGTGTACCTGAAGCAGATG–3' (400 μM) was used (*Löhr et al., 2009*; *Cheng and Wingert, 2015*). RT-PCR was completed to determine efficacy of the *sim1a* splice MO knockdown as previously described (*Marra and Wingert, 2016*). To complete RT-PCR the following primers were used: *ppargc1a*-FWD 5'–AATGCCAGTGATCAGAGCTGTCCTT–3', ppargc1a-RVS 5'–CAGCTCAGTGCAGGGACGTCTCATG–3', *sim1a*-FWD 5'–GAATCTTGGGGCCATGTGAGTCGAACGACTTCACTGG–3', *sim1a*-RVS 5'–GTACAGGATTTTCCCATCAGGAGCCACCACAAAGATG–3'.

## Quantitative real-time PCR

Groups of 25–30 bezafibrate treated and vehicle control zebrafish were pooled with their respective group at the 28 ss. RNA was extracted using TRIZOL (Ambion) following the manufacturer instructions. cDNA was generated by qScript cDNA SuperMix (QuantaBio). qRT-PCR reactions were completed using PerfeCTa SYBR Green SuperMix with ROX (QuantaBio). To target *ppargc1a*, 100 ng of cDNA was optimal. For 18S control 1 ng was optimal. Primers used to amplify *ppargc1a* were forward 5'–AATGCCAGTGATCAGAGCTGTCCTT–3' and reverse 5'–GTTCTGTGCCTTGCCACCTGGGTAT–3'. To target 18S the primers were as follows: forward 5'–TCGGCTACCACATCCAAGGAAGGCAGC–3' and reverse 5'–TTGCTGGAATTACCGCGGCTGCTGGCA–3'. The AB StepOnePlus quantitative real time PCR machine program was: holding stage 2 min at 50°C, holding stage for 10 min at 95°C, then cDNA was amplified during 40 cycles, alternating between 15 s at 95°C to denature the cDNA and 1 min at 62°C for primer annealing and product extension. Data were recorded after each cycle to obtain the Ct values. Three technical replicates were completed for each of the three biological replicates for both treatments with the median Ct value normalized to control. Delta delta Ct was used for data analysis with 18S as a reference gene and the results calculated as relative expression change relative to DMSO control. For statistical analysis a Student's t test was performed using the delta Ct values obtained after normalization to the 18S reference gene.

## cRNA synthesis, and microinjections, rescue studies

The zebrafish *ppargc1a* ORF was cloned in to a pUC57 vector flanked by a 5' KOZAK sequence, a Cla1 restriction enzyme site, and a SP6 promoter region, and on the 3' by a series of STOP codons, a SV40 poly A tail, a Not1 restriction enzyme site, and a t7 promoter region. *ppargc1a* cRNA was generated by linearizing with Not1 and sp6 run off with the mMESSAGE mMACHINE SP6 Transcription kit (Ambion). cRNA was injected into WT and *ppargc1a*$^{sa13186}$ mutants at the one-cell stage at a concentration of 900 pg. Rescue studies were completed by performing WISH on injected *ppargc1a*$^{sa13186}$ mutants, then samples were imaged and genotyped. A minimum of three samples for each genotype was used to calculate segment phenotypes.

## Image acquisition and quantification of phenotypes

WISH images were acquired using a Nikon Eclipse Ni with a DS-Fi2 camera. FISH and immunofluorescence images were acquired using a Nikon C2 confocal microscope. Segment phenotypes were quantified using the Nikon Elements imaging software polyline tool. Unless otherwise stated, a minimum of three representative samples from each biological replicate were imaged and measured. Experiments were completed in triplicate. From these measurements, an average and standard deviation (SD) were calculated, and unpaired t-tests or one-way ANOVA tests were completed to compare control and experimental measurements.

## Acknowledgements

NIH Grant R01DK100237 to RAW supported this work. We are grateful to Elizabeth and Michael Gallagher for a generous gift to the University of Notre Dame on behalf of their family for the support of stem cell research. The funders had no role in the study design, data collection and analysis, decision to publish, or manuscript preparation. We thank the staffs of the Department of Biological Sciences and the Center for Zebrafish Research at the University of Notre Dame for their dedication and care of our zebrafish aquarium. We thank ED for contributions to preliminary investigations on the effects of bezafibrate on zebrafish nephron development. We also extend special thanks to LG for assistance with analytical tools. Finally, we thank all the current and previous members of our lab for their support, discussions, and insights about this work.

## Additional information

### Funding

| Funder | Grant reference number | Author |
| --- | --- | --- |
| National Institutes of Health | R01DK100237 | Rebecca A Wingert |

The funders had no role in study design, data collection and interpretation, or the decision to submit the work for publication.

### Author contributions

Joseph M Chambers, Conceptualization, Data curation, Formal analysis, Validation, Investigation, Visualization, Methodology, Writing—original draft, Project administration, Writing—review and editing; Shahram Jevin Poureetezadi, Conceptualization, Data curation, Formal analysis, Investigation, Methodology; Amanda Addiego, Data curation, Formal analysis, Validation, Investigation, Visualization; Manuela Lahne, Data curation, Formal analysis, Investigation, Methodology, Writing—review and editing; Rebecca A Wingert, Conceptualization, Data curation, Formal analysis, Supervision, Funding acquisition, Validation, Investigation, Visualization, Methodology, Writing—original draft, Project administration, Writing—review and editing

### Author ORCIDs

Rebecca A Wingert (iD) http://orcid.org/0000-0003-3133-7549

### Ethics

Animal experimentation: Zebrafish were maintained in the Center for Zebrafish Research at the University of Notre Dame. All studies were performed with approval of the University of Notre Dame Institutional Animal Care and Use Committee (IACUC), under protocol numbers 13-021 and 16-025.

### Decision letter and Author response

Decision letter https://doi.org/10.7554/eLife.40266.026
Author response https://doi.org/10.7554/eLife.40266.027

## Additional files

### Supplementary files

• Transparent reporting form
DOI: https://doi.org/10.7554/eLife.40266.023

### Data availability

All data generated or analysed during this study are included in the manuscript and supporting files.

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
