## [Decision Letter]

Thank you for submitting your article "*ppargc1a* controls nephron segmentation during zebrafish embryonic kidney ontogeny" for consideration by *eLife*. Your article has been reviewed by two peer reviewers, and the evaluation has been overseen by Tanya Whitfield as the Reviewing Editor and Sean Morrison as the Senior Editor. The following individual involved in review of your submission has agreed to reveal his identity: Dirk Meyer (Reviewer #2).

The reviewers have discussed the reviews with one another and the Reviewing Editor has drafted this decision to help you prepare a revised submission.

Summary:

The manuscript by Chambers et al. describes a novel role of the transcriptional coactivator Ppargc1a/PGC-1α in embryonic zebrafish pronephros patterning and segmentation. The experiments build on results of an earlier small compound screen from the same lab in which PPAR agonists were found to interfere with nephron patterning. In this manuscript experiments are described that present *ppargc1a* as a promising candidate involved in mediating the underlying patterning activities. By combining genetic, morpholino and RNA-injection based functional approaches with WISH-based phenotype analyses the authors show that *ppargc1a* is involved in nephron pattering and they reveal evidence for interactions between *ppargc1a, tbx2b* and *sim1a* in the proper positioning of distal pronephric segment boundaries.

Essential revisions:

Both reviewers are positive about the work, but both suggest revisions, some of which are overlapping.

1) Please use qRT-PCR to quantify the expression changes detailed in the manuscript (a suggestion of both reviewers).

2) Please revise the arguments about epistasis so that they refer to the comparison of double and single mutant phenotypes and not expression analysis in a single mutant, as suggested by reviewer 1.

3) Provide more data on the initial screen (reviewer 2).

4) Include a more comprehensive study of cell proliferation and cell death (reviewer 2).

5) Test for cross-repressive interactions by RNA overexpression (reviewer 2).

For further information, the full reviews are appended below.

*Reviewer #1:*

This study offers a significant advance in delineating the genetic hierarchies that regulate nephron segment patterning during zebrafish development. The authors follow up on their earlier work, in which through unbiased small molecule screens, they identified PPAR signaling as a candidate pathway for regulating this process. Here, the authors show that *ppargca1* encoding PPAR coactivator is expressed in intermediate mesoderm in early zebrafish embryos and this expression is dynamic in the course of kidney development. Through a combination of various single and compound loss and gain of function approaches, the authors place *ppargc1a* upstream of *tbx2b* in promoting distal segment formation and discover mutually antagonistic regulatory relationship between *ppargc1a* and *sim1a* in establishment of the proximal straight tubule (PST) segment boundaries. However, because loss of PST markers is suppressed by *ppargc1a* deficiency, with the compound deficiency presenting with a normal PST pattern, this work implies that a redundant patterning mechanism must be at work. Given the conservation of kidney segment patterning processes across vertebrates, these relationships are likely to inform and be conserved in other animals. The work is in general compelling and data very well and clearly presented. Therefore, this work should be of interest to the broad developmental and regenerative biology community. However, some of the conclusions need additional experimental support and some interpretations of the results need to be revised, before the manuscript becomes suitable for publication.

– The conclusion that bezafibrate treatment affects *ppargc1a* expression is not compelling and requires further experimental analysis. Currently, expression of *ppargc1a* is analyzed in the drug treated embryos during late segmentation (28S – the stage is not provided on the Figure 2 or in the text) and reduced expression domain is observed. However, by this stage as shown in Figure 1, *ppargc1a* expression is limited to the cells that express the distal segment markers. Therefore, that *ppargc1a* expression domain is reduced could simply reflect that DL is reduced what is the key finding of the manuscript. The authors also note that expression level of *ppargc1a* RNA appears reduced, but WISH is not a quantitative method. Therefore, it would be important to analyze *ppargc1a* expression at 15 somite stage when the authors show it is expressed in the entire domain of cells expressing *pax2a*, or intermediate mesoderm. At this stage qRT PCR should also be performed.

– A conclusion is reached that *ppargc1a* is epistatic to *tbx2b* based on the observations that in *ppargc1a* deficient embryos expression of *tbx2b* is impaired but expression of *ppargc1a* is not altered in *tbx2b* deficient embryos. Epistatic relationships are based on combining and comparing contrasting loss or gain of function phenotypes. Whereas the two LOF phenotypes are similar, gain of *tbx2b* function/expression masks the loss of *ppargc1a*, supporting the notion that *tbx2b* is epistatic to *ppargc1a*, but *ppargc1a* acts upstream of *tbx2b*.

– Likewise, that domain of *ppargc1a* expression in renal progenitors was significantly increased in length, is interpreted that "*sim1a* was probably epistatic to *ppargc1a*". Again these are not experiments to test epistasis. The authors do perform such experiment by creating a compound loss of function between these two genes, which have contrasting LOF phenotypes, with PST fates being lost in *sim1a* deficient embryos and expanded in *ppargc1a* mutants. And because *ppargc1a;sim1a* doubly deficient embryos show normal PST segment, it is *ppargc1a* that is epistatic to *sim1a*. Even though epistasis is not complete, consistent not only with *pparg1a* negatively regulating *sim1a* expression but also vice versa. The conclusions about epistatic relationships need to be revised both in the Results and Discussion sections.

*Reviewer #2:*

The authors convincingly show that PPAR agonist treatment and genetic loss of *ppargc1a* result in very similar minor displacements of distal segment boundaries and that the dynamic nephric-specific expression of *ppargc1* correlates with a cell-autonomous role during nephron patterning. They also show that shortening of the DL segment in *ppargc1* mutants is associated with a shortened expression of the DL-fate determining factors *tbx2b* and that *tbx2b* RNA injection can restore DL positioning in *ppargc1* mutants. Further they reveal genetic evidence for cross-repressive interactions between *ppargc1a* and *sim1* in positioning the distal extension of the PST segment. The design of the experiments is clear and approaches are straightforward. However, some important questions remain unanswered and some of the results may require further analyses.

Subsection “Bioactive small molecule chemical genetic screen reveals that alteration of PPAR signaling leads to changes in embryonic nephron segmentation”: More data from the initial screen concerning 'compounds that are known to alter activity of PPAR signaling' should be provided. These data might shed some light on the somehow contradictory and currently poorly explained observations that PPAR agonist treatment and loss of *ppargc1* cause very similar phenotypes.

Subsection “*ppargc1a* is necessary for proper formation of proximal and distal segment boundaries”, last paragraph: Benzafibrate treatments restricted to the time window of *ppargc1a* expression (>8 somite stage) rather than starting at 5hp should be used to better define the correlation between activation of PPAR signaling and *ppargc1a* expression/function. In case of a direct role of benzafibrate in repressing *ppargc1* expression it should also be possible to rescue the phenotype by *ppargc1* RNA-injection. Alternatively, the distal shift of *ppargc1* expression in benzafibrate treated embryos may be caused by an earlier patterning defect that cannot be rescued by RNA injection.

“Further, the *ppargc1a* signal intensity appeared…”: Quantitative analyses such as RT-qPCR should be used to confirm the suggested general reduction.

“…whether either of these changes were associated with regional fluctuations in cell birth or death.”: To answer this relevant and interesting question, it would have been necessary to study proliferation and cell death in relation to the segment/region specific markers *trpm7* and *slc12a3* rather than using the pan-nephric marker *cdh17*.

Subsection “*ppargc1a* regulates PST boundary formation through a reciprocally antagonistic relationship with the *sim1a* transcription factor”: The genetic cross-repressive interaction between *ppargc1a* and *sim1a* is interesting. However, to better understand the underlying molecular mechanisms it would be important to further explore whether cross-repressive interactions can be seen when using overexpression of *ppargc1a* and *sim1a* RNA. In this context the authors should also explain why embryo-wide overexpression of their RNAs appears to cause very restricted phenotypes.

---

## [Author Response]

Essential revisions:Both reviewers are positive about the work, but both suggest revisions, some of which are overlapping.1) Please use qRT-PCR to quantify the expression changes detailed in the manuscript (a suggestion of both reviewers).

As suggested by both of the reviewers, we have used qRT-PCR to quantify *ppargc1a* transcripts in response to treatment with bezafibrate, which is provided in revised Figure 2G.

2) Please revise the arguments about epistasis so that they refer to the comparison of double and single mutant phenotypes and not expression analysis in a single mutant, as suggested by reviewer 1.

The arguments about epistasis have been revised so that they refer to comparison of double and single mutant phenotypes.

3) Provide more data on the initial screen (reviewer 2).

The manuscript has been revised to provide more data on the initial screen, which is now located in Figure 1—figure supplement 1.

4) Include a more comprehensive study of cell proliferation and cell death (reviewer 2).

As suggested we performed comprehensive studies of cell proliferation and death at the 20 ss in the PST and DL, which are the two affected segments in *ppargc1a* deficient embryos, using combined fluorescent in situ hybridization and immunofluorescence techniques. We found that there was no significant difference in proliferation or death in either segment. This is consistent with the conclusion that changes in cellular turnover do not cause the PST and DL segment boundary changes that occur in *ppargc1a* deficient embryos.

5) Test for cross-repressive interactions by RNA overexpression (reviewer 2).

We performed RNA overexpression studies to assess the effect of *sim1a* RNA on the domain of *ppargc1a* expression in the kidney, and vice versa, and to evaluate the coincident development of the PST segment (Figure 5). Overexpression of *sim1a* was associated with a statistically significant reduction in the *ppargc1a* domain (Figure 5A, 5D) and an enlarged PST segment (Figure 5C, 5F). Overexpression of *ppargc1a* led to a statistically significant reduction in the *sim1a* domain (Figure 5B, 5E) at 20 ss and a shortened PST segment (Figure 5C, 5F) in addition to a decrease in the number of *sim1a*^+^ cells at 28 ss (Figure 5—figure supplement 2). These results with RNA overexpression provide further support for the conclusion that there are cross-repressive interactions between *sim1a* and *ppargc1a* during pronephros ontogeny.

For further information, the full reviews are appended below.Reviewer #1:[…]– The conclusion that bezafibrate treatment affects ppargc1a expression is not compelling and requires further experimental analysis. Currently, expression of ppargc1a is analyzed in the drug treated embryos during late segmentation (28S – the stage is not provided on the Figure 2 or in the text) and reduced expression domain is observed. However, by this stage as shown in Figure 1, ppargc1a expression is limited to the cells that express the distal segment markers. Therefore, that ppargc1a expression domain is reduced could simply reflect that DL is reduced what is the key finding of the manuscript. The authors also note that expression level of ppargc1a RNA appears reduced, but WISH is not a quantitative method. Therefore, it would be important to analyze ppargc1a expression at 15 somite stage when the authors show it is expressed in the entire domain of cells expressing pax2a, or intermediate mesoderm. At this stage qRT PCR should also be performed.

We performed further experimental analyses to assess how bezafibrate treatment affects *ppargc1a* expression. First, we performed studies to investigate the time window of bezafibrate treatment in order to better define the correlation between activation of PPAR signaling and *ppargc1a* expression. Bezafibrate was added at the 4, 5, 6, 8, 9, and 10 hpf stage as well as the 5 ss and 10 ss, and embryos were incubated until the 28 ss, at which time they were fixed and *ppargc1a* expression was assessed by WISH (Figure 2—figure supplement 5). Bezafibrate addition at the 4, 5, 6, 8, or 9 hpf time points led to statistically significant decreases in the *ppargc1a* domain length compared to control embryos; further the effect of bezafibrate addition at these time points was not statistically different when compared by ANOVA (Figure 2—figure supplement 5A, B). Thus, we next added bezafibrate at the 5 hpf stage and fixed embryos at the 15 ss to analyze *ppargc1a* expression. Interestingly, we found no difference in *ppargc1a* expression between bezafibrate treated and control embryos (Figure 2—figure supplement 5C). Finally, we conducted qRT-PCR analysis on pools of embryos treated with bezafibrate or DMSO vehicle control from the 5 hpf time point until the 28 ss. Although this time window of bezafibrate exposure was associated with a reduced absolute domain length of *ppargc1a* expression in the pronephros (Figure 2E, 2F), the qRT-PCR analysis revealed that the relative mRNA levels were not significantly different between the groups (Figure 2E). Furthermore, we found that *ppargc1a* RNA overexpression was not sufficient to rescue DL development in bezafibrate treated embryos (Figure 2—figure supplement 5D, 5E). Taken together, these results are consistent with the notion that the *ppargc1a* expression domain is reduced in bezafibrate treated embryos because the DL is reduced, and not specifically due to the loss of *ppargc1a* activity. We have revised the manuscript to reflect this conclusion.

Additionally, we have addressed the reviewer’s comment about missing information in the legend to Figure 2, to clarify that the embryonic stage of the representative images is the 28 ss.

– A conclusion is reached that ppargc1a is epistatic to tbx2b based on the observations that in ppargc1a deficient embryos expression of tbx2b is impaired but expression of ppargc1a is not altered in tbx2b deficient embryos. Epistatic relationships are based on combining and comparing contrasting loss or gain of function phenotypes. Whereas the two LOF phenotypes are similar, gain of tbx2b function/expression masks the loss of ppargc1a, supporting the notion that tbx2b is epistatic to ppargc1a, but ppargc1a acts upstream of tbx2b.

We have revised the manuscript to correct the conclusions about epistatic relationships.

– Likewise, that domain of ppargc1a expression in renal progenitors was significantly increased in length, is interpreted that "sim1a was probably epistatic to ppargc1a". Again these are not experiments to test epistasis. The authors do perform such experiment by creating a compound loss of function between these two genes, which have contrasting LOF phenotypes, with PST fates being lost in sim1a deficient embryos and expanded in ppargc1a mutants. And because ppargc1a;sim1a doubly deficient embryos show normal PST segment, it is ppargc1a that is epistatic to sim1a. Even though epistasis is not complete, consistent not only with pparg1a negatively regulating sim1a expression but also vice versa. The conclusions about epistatic relationships need to be revised both in the Results and Discussion sections.

We have revised the manuscript to correct the conclusions about epistatic relationships.

Reviewer #2:[…]Subsection “Bioactive small molecule chemical genetic screen reveals that alteration of PPAR signaling leads to changes in embryonic nephron segmentation”: More data from the initial screen concerning 'compounds that are known to alter activity of PPAR signaling' should be provided. These data might shed some light on the somehow contradictory and currently poorly explained observations that PPAR agonist treatment and loss of ppargc1 cause very similar phenotypes.

In the PPAR literature, other laboratories have reported both increases and decreases of *ppargc1a* after treatment with PPAR agonists and antagonists (Pardo et al., 2011; Liao et al., 2010; Sanoudou et al., 2010; Goto et al., 2017; Wang and Moraes, 2011). This suggests to us that different cells respond differently to PPAR pathway modulation with respect to *ppargc1a/PGC1a* expression.

We provide additional data from the initial screen in the revised Figure 1—figure supplement 1. Along with bezafibrate, a PPAR alpha agonist, there were 2 PPAR gamma antagonists identified as screen hits, compounds known as BADGE and GW-9662. Both PPAR gamma antagonists led to an expanded DE, and GW-9662 also led to an expanded PCT based on double WISH (for the DE marker *slc12a1* and the PCT marker *slc20a1a*). However, the screen did not directly assess either the PST or DL segments.

Subsection “ppargc1a is necessary for proper formation of proximal and distal segment boundaries”, last paragraph: Benzafibrate treatments restricted to the time window of ppargc1a expression (>8 somite stage) rather than starting at 5hp should be used to better define the correlation between activation of PPAR signaling and ppargc1a expression/function. In case of a direct role of benzafibrate in repressing ppargc1 expression it should also be possible to rescue the phenotype by ppargc1 RNA-injection. Alternatively, the distal shift of ppargc1 expression in benzafibrate treated embryos may be caused by an earlier patterning defect that cannot be rescued by RNA injection.

Please see our response to reviewer #1, comment 1.

“Further, the ppargc1a signal intensity appeared…”: Quantitative analyses such as RT-qPCR should be used to confirm the suggested general reduction.

As noted above, qRT-PCR was performed to measure *ppargc1a* signal in bezafibrate treated embryos. We have revised the manuscript accordingly.

“…whether either of these changes were associated with regional fluctuations in cell birth or death.”: To answer this relevant and interesting question, it would have been necessary to study proliferation and cell death in relation to the segment/region specific markers trpm7 and slc12a3 rather than using the pan-nephric marker cdh17?

As suggested we performed cell proliferation and death immunofluorescence studies in combination with fluorescent in situ hybridization for the segment-specific markers *trpm7* (PST) and *slc12a3* (DL), which is now presented in revised Figure 3. Wild-type and *ppargc1a* deficient embryos had no statistically significant difference in the numbers of proliferating or dying cells in either the PST or the DL.

Subsection “ppargc1a regulates PST boundary formation through a reciprocally antagonistic relationship with the sim1a transcription factor”: The genetic cross-repressive interaction between ppargc1a and sim1a is interesting. However, to better understand the underlying molecular mechanisms it would be important to further explore whether cross-repressive interactions can be seen when using overexpression of ppargc1a and sim1a RNA. In this context the authors should also explain why embryo-wide overexpression of their RNAs appears to cause very restricted phenotypes.

Please see our response for Essential revisions, point #5.

Embryo-wide overexpression of these RNAs may cause restricted phenotypes because only certain tissues are competent to respond to altered levels of these transcripts. It is also possible that embryo-wide effects would be discernible at higher mRNA dosages, but the present work has not exhaustively chronicled the dosages and resultant phenotypes of each RNA through overexpression studies.